# Biocatalytic Synthesis of Coumarin *S*-Glycosides: Towards Non-Cytotoxic Probes for Biomedical Imaging and Sensing

**DOI:** 10.3390/molecules29061322

**Published:** 2024-03-16

**Authors:** Nastassja Burrini, Arnaud Pâris, Guillaume Collet, Pierre Lafite, Richard Daniellou

**Affiliations:** 1Institut de Chimie Organique et Analytique, University of Orléans CNRS UMR7311, F-45067 Orléans, France; nastassja.burrini@univ-orleans.fr (N.B.); arnaud.paris@univ-orleans.fr (A.P.); 2Chaire de Cosmétologie, AgroParisTech, 10 rue Léonard de Vinci, F-45100 Orléans, France; guillaume.collet@agroparistech.fr; 3Université Paris-Saclay, INRAE, AgroParisTech, Micalis Institute, F-78350 Jouy-en-Josas, France

**Keywords:** biocatalyst, *S*-glycoside, mercapto-coumarin, fluorescence

## Abstract

This study unveils an innovative method for synthesizing coumarin *S*-glycosides, employing original biocatalysts able to graft diverse carbohydrate structures onto 7-mercapto-4-methyl-coumarin in one-pot reactions. The fluorescence properties of the generated thio-derivatives were assessed, providing valuable insights into their potential applications in biological imaging or sensing. In addition, the synthesized compounds exhibited no cytotoxicity across various human cell lines. This research presents a promising avenue for the development of coumarin *S*-glycosides, paving the way for their application in diverse biomedical research areas.

## 1. Introduction

Glycosides, a prominent class of natural compounds, play a pivotal role in numerous biological processes and hold substantial potential for therapeutic applications [1,2,3]. Among these, *O*-glycosides and *S*-glycosides stand out as essential subclasses, distinguished by the nature of the linkage connecting the sugar moiety to aglycone. *O*-glycosides, characterized by the attachment of the glycosidic linkage to an oxygen atom, constitute a vast and versatile class of compounds. Abundantly present in natural products and plant secondary metabolites, *O*-glycosides significantly contribute to the structural complexity and functionality of these bioactive molecules. Understanding the stereo- and regio-chemistry of *O*-glycosidic linkages is crucial for elucidating their biological roles and harnessing their applications in drug discovery and development. Simultaneously, the exploration of *S*-glycosides has emerged as an exciting field in glycoside chemistry. These structures, wherein the conventional anomeric oxygen is replaced by a sulfur atom, show remarkable physicochemical properties. Of particular interest is their heightened stability towards enzymatic and acid/base hydrolyses when compared to their *O*-glycoside counterparts [4]. Consequently, *S*-glycosides have proven to be useful tools in chemical synthesis and functioning as stable ligands essential for the crystallographic analysis of protein–ligand complex structures [5]. Moreover they play a pivotal role in the development of carbohydrate-based vaccine antigens [6,7] and act as substrate analogues or inhibitors of Glycosides Hydrolases (GHs) involved in many diseases, including cancer, lysosomal storage disorder, and viral and bacterial infections [8]. In the domain of organic chemistry, the evaluation of these derivatives has led to the development of intricate synthesis methodologies [9,10] and metal-catalyzed reactions [11,12,13], necessitating elaborated protection/deprotection procedures and precise control over the configuration of the anomeric carbon. Enzymes have emerged as potent synthetic tools facilitating the environmentally sustainable generation of diverse compound classes. The employment of non-organic solvents and mild experimental conditions and the intrinsic high regio- or stereo-specificity inherent to biocatalyzed reactions have considerably enhanced the utility of enzymes in transformation processes, ranging from laboratory-scale to industrial context [14]. Starting from these interesting results, in 2003, Withers et al. presented the first generation of thioglycoligases (Abg E170A from *Aspergillus* sp. and Man2A E249A from *Cellulomonas fimi*). These enzymes demonstrated the capability to catalyze the formation of *S*-glycosidic bonds [15]. Derived from retaining GHs, their acid/base residue essential for the hydrolytic activity has been substituted via site-directed mutagenesis with a non-catalytically active amino acid, such as alanine (A), glutamine (Q), or asparagine (N). Following a comparable strategy, our research team has developed a repertoire of thioglycoligases through the selective modification of the active site of several GHs, derived from the thermophilic organism *Dictyoglomus thermophilum* [4,16,17,18]. These mutants, characterized by a drastic reduction in their hydrolytic activities due to the replacement of the catalytic residue with either alanine or glutamine, can efficiently catalyze the binding of a specific carbohydrate molecule onto an aromatic thiol, resulting in the synthesis of *S*-glycoside derivatives with a high yield.

In the context of this study, 7-mercapto-4-methylcoumarin (**7-MC**) has been identified as a valuable sugar acceptor [17,19,20] (Figure 1). Within the expansive family of coumarins, this compound exhibits the common structure with the benzene ring fused to an α-pyrone moiety. Notably, the distinctive sulfur-side chain at position 7 contributes to the compound’s reactivity, offering a catalyst for derivatization, enabling precise customization for targeted applications or desired biological activities [21]. Despite the relatively lower spotlight compared to its *O*-counterpart, 4-methylumbelliferone (**4-MUB**), **7-MC** proves to be chemically and bioactively promising in the field of biological and material sciences. Recognized as a frontier substrate, it serves as a versatile platform to synthesize a variety of bioactive compounds [22,23]. Moreover, its efficacy as a matrix for analyzing small molecular compounds is acknowledged [24]. The versatility of **7-MC** is highlighted across diverse applications, as evidenced by its role as a reporter molecule [25], involvement in thiol interactions at the nanoparticle surface [26], serving as Raman reporter [27,28,29], proving to be an excellent substrate for fluorescence spectroscopy [30], demonstrating photodimerizable and healable properties [31], and functioning as fluorescent dye [32,33]. Additionally, it acts as a probe molecule on gold-coated silicon nanowires [34] and excels as a fluorophore [35,36,37], making it particularly valuable in biological and chemical sensing applications. This compound serves as an effective tool for the detection and quantification of various substances, including biomolecules, or to evaluate the activity of specific enzymes belonging to the family of GHs [4,38]. Its fluorescence emission is responsive to specific environmental conditions or interactions with targets, rendering it a versatile asset for sensing and imaging purposes.

In this article, we present the efficient biocatalyzed synthesis of diverse *S*-glycosidic coumarins, along with their cytotoxic evaluation and exploration of their fluorescent properties as promising fluorescent probes.

## 2. Results and Discussion

### 2.1. Biocatalyzed Synthesis of S-Glycosidic Coumarins

To ensure the catalytic efficacy of the mutated GHs, it is necessary to employ glycosyl donors and acceptors compatible with the double-displacement mechanism inherent in configuration-retaining glycosides [15].

The activation of sugars is essential for conducting the enzymatic glycosylation step efficiently. For this reason, a panel of commercially available *p*-nitrophenyl-d-glycopyranosides (*p*NP-glycosides) has been selected as sugar donors (Figure 1). To prevent the formation of disulfide bonds, the reactions were carried out in the presence of a large excess of dithiothreitol (DTT, 10 equiv.) and followed by HPLC. The resulting products were purified by semi-preparative HPLC and characterized by NMR and HRMS, as reported in the Appendix A and in Table 1.

The synthesis of *S*-glycosides poses significant challenges in chemical synthesis, often requiring multiple protection/deprotection steps, intricate stereochemical control, and extensive purification efforts [39,40]. However, an alternative approach utilizing enzymatic catalysis with engineered GHs presents an attractive option due to its ability to yield products with a pure anomeric configuration. In this study, several thioglycoligases were used as catalysts to glycosylate **7-MC** in a one-pot reaction.

Compound **S-1** was obtained following a previously established procedure [17]. The aromatic acceptor was efficiently glycosylated by DtGlcAE396Q mutant, using *p*NP-GlcA as the glucuronide donor, resulting in a 75% conversion, as determined by HPLC analysis. The ^1^H-NMR analysis shows a chemical shift (δ 4.97 ppm) and a coupling constant (*J* = 9.1 Hz), confirming the β-configuration of the aryl-glucuronide.

As compound **S-1**, the other thioglycosides, **S-2** to **S-6**, bearing, respectively, β-d-xylopyranosyl, β-d-glucopyranosyl, β-d-galactopyranosyl, β-d-fucopyranosyl, and α-d-galactopyranosyl moieties, were also synthetized in a one-step process and the anomeric configurations were assessed by ^1^H-NMR spectroscopy.

In this study, we used the efficacy of several mutated GHs, with each exhibiting selective affinity towards specific glycosides, in catalyzing the thioligation reaction between the activated sugar substrate and 7-MC. This approach bypasses the need for laborious protection/deprotection protocols typically encountered in chemical synthesis, thereby simplifying the formation of *S*-glycosides with the desired configuration [11,41].

Our research underscores the potential of enzymatic glycosylation as a robust method for synthetizing *S*-derivatives with remarkable efficiency [19]. Through the fusion of engineered GHs’ selectivity and efficiency with the simplicity and convenience of one-pot reactions, we have developed a versatile and practical approach for grafting diverse glycoside coumarins. Notably, our work marks the inaugural enzymatic synthesis of **S-2**, **S-4** to **S-6**, a significant milestone in glycoscience. This innovative methodology not only facilitates precise control over stereochemistry and molecular structure but also represents a pioneering leap forward in the field.

### 2.2. Cytotoxic Activity

To use these *S*-glycosidic compounds in cellulo, the potential impact of the glycosidic linkage nature on the anticancer activity was assessed. The *S*-glycoside derivatives and the 7-MC were tested against four human cancer cell lines and a healthy one, namely A549 (lung carcinoma), HS-683 (glioma cancer), MCF-7 (breast adenocarcinoma), SK-MEL-28 (melanoma skin cancer), and HaCaT (keratinocyte normal skin), using MTT (thiazolyl blue tetrazolium bromide, Sigma-Aldrich, St. Louis, MO, USA) assay, as previously described in the literature [42,43,44]. The selected reference drugs were 5-fluorouracil (5-FU) and etoposide. The results listed in Table 2 (IC50 value, defined as the concentration, inducing a 50% decrease in cell growth after 3 days of incubation) show that most coumarin-containing compounds display no inhibitory activities against the cell-line panel tested (>1000 µM). The only exception is **S-2**, which shows a moderate cytotoxic activity against the lung carcinoma cell line A-549, with an IC50 value of 74 µM. 

This finding is particularly noteworthy, considering the well-documented cytotoxic properties associated with umbelliferone and its derivatives, which share the coumarin structural framework [45,46,47]. The discrepancy between our results and prior research underscores the importance of evaluating the cytotoxicity of structurally modified compounds to optimize their pharmacological properties. 

Given their non-cytotoxic nature, *S*-glycosyl coumarins hold promise for alternative biomedical applications. These include their potential use as imaging agents, incorporation into drug delivery systems for the safe transportation of therapeutic agents to targeted cells or tissues, and utilization as biochemical probes to study biochemical processes. In fact, while our study primarily focused on their non-cytotoxic properties, further research may uncover additional therapeutic applications for *S*-glycosyl coumarins, leveraging their unique chemical properties, such as anti-inflammatory or antioxidant effects. 

### 2.3. Optical Spectroscopy

UV–visible spectral analysis was conducted using dimethylformamide (DMF) as the initial solvent at a concentration of 10 mM to ensure optimal solubility of the compounds. Subsequently, the analysis was carried out under two distinct conditions: firstly, in DMF, at a final concentration of 0.1 mM; and secondly, in a mixture PBS:DMF (99:1%), at the same final concentration of 0.1 mM. These two experimental setups provided valuable insights into the spectral characteristics of the compounds in both a pure organic solvent and a solvent system closely resembling physiological conditions. The maxima absorption wavelengths are compiled in Table 3. These bands were attributed to transitions within the benzene and pyrone moieties, considered together as a single chromophore. In the UV region, the absorbance spectrum of **4-MUB** displayed a broad peak centered at λ_max_ = 322 nm, using DMF as solvent. Similarly, **7-MC** and **S-1** to **S-6** compounds also absorb in this UV range (λ_max_ = 328–330 nm). However, the extinction coefficients for *S*-glycosides exhibit variability. In the organic solvent, these values are found to be higher compared to those calculated for **4-MUB** and **7-MC**. Conversely, in PBS, the extinction coefficients for *S*-glycosides are lower compared to the parent thiol **7-MC**. This highlights the profound impact of glycosylation on absorption properties, as well as the importance of solvent choice. 

Furthermore, an increase in solvent polarity, transitioning from DMF to PBS, induced shifts in absorption spectra, either redshifts (**7-MC** and **4-MUB**) or blueshifts (all *S*-glycosides). This phenomenon implies a significant disparity between the polarity of the ground and excited states [48]. The introduction of a -SH substitution into the coumarin core notably perturbed the spectrum, while variations in pH strongly influenced the absorption properties of **7-MC** [49]. To mitigate potential disulfide bridge formation, spectra of **7-MC** were recorded with the addition of 10 mM DTT to the solution. 

**Table 3 molecules-29-01322-t003:** Spectroscopic properties of **4-MUB**, **7-MC**, and *S*-derivatives in organic solvent and PBS: absorption, emission, and excitation fluorescence wavelengths maxima (λ_max_); molar absorption coefficient (ε_max_); and relative quantum yield, Φ.

Compound	Absorbance				Fluorescence					
	λ_max abs_ (nm)		ε (10^3^ M^−1^ cm^−1^)		λ_max ex_ (nm)		λ_max em_ (nm)		Φ	
	DMF	PBS	DMF	PBS	DMF	PBS	DMF	PBS	DMF	PBS
**4-MUB**	322	324	17.0	20.2	320	333	380	448	0.081 ^a^	0.293
**7-MC ^b^**	328	368	10.4	28.9	334	321	392	448	0.009	0.020
**S-1**	330	326	26.7	23.3	332	332	390	402	0.182	0.235
**S-2**	330	324	19.5	26.0	332	326	386	402	0.186	0.181
**S-3**	330	326	18.6	20.8	332	332	386	402	0.198	0.242
**S-4**	330	326	22.3	18.4	334	332	388	402	0.223	0.271
**S-5**	330	326	22.3	10.4	334	326	386	402	0.183	0.228
**S-6**	330	326	22.3	19.8	334	332	388	416	0.223	0.185

^a^ From ref. [50]. ^b^ In presence of 10 mM dithiothreitol (DTT).

All compounds displayed similar fluorescence excitation maxima (320–333 nm) and emission maxima (380–392 nm) wavelengths in DMF (Table 3, Figure 2, and Appendix A). In PBS solvent, this homogeneity in the latter disappeared, as the unglycosylated **4-MUB** and **7-MC** exhibited fluorescence emission maxima at 448 nm. On the other hand, the glycosylation of **S-1** to **S-6** led to similar fluorescence emission maxima at 402–416 nm wavelengths (Table 3 and Appendix A).

Quantum yields (relative to **4-MUB** in DMF) [50] were also determined for all *S*-glycosides, and they revealed that grafting a glycoside onto **7-MC** gave a higher yield. Indeed, when using either DMF or PBS as the solvent, compounds **S-1** to **S-6** exhibited a 10-fold increase in their quantum yield compared to **7-MC**. The values determined were close to those observed for the **4-MUB** fluorophore.

The choice of the solvent plays a crucial role in dictating the fluorescence behavior of the investigated compounds. Notably, when comparing their behavior in PBS to DMF, we observed a consistent increase in emission intensity for the *S*-derivatives in PBS, except for **S-2** and **S-6**. This discrepancy primarily arises from the different polarity and hydrogen-bonding capabilities of the two solvents. PBS is renowned for its high polarity and the presence of phosphate ions, which can readily engage in hydrogen-bonding interactions with the functional groups of the *S*-glycosides. These interactions facilitate the stabilization of the excited state, consequently leading to heightened fluorescence emission. Moreover, the polar environment of PBS may shield the molecules from non-radiative decay processes, further enhancing fluorescence intensity. Conversely, the organic nature of DMF introduces distinct dynamics to the excited state relaxation pathways of the *S*-glycosides. Being a less polar solvent, DMF may exert a weaker influence on the stabilization of the excited states compared to PBS. Additionally, the formation of hydrogen-bonding interactions, different from those present in PBS, might alter the energetic landscape of the excited states, potentially resulting in lower fluorescence emission and quantum yields. Furthermore, microenvironment effects within each solvent could also contribute to the observed differences in fluorescence behavior. Variations in solute–solvent interactions, solvent accessibility to the chromophore, and local concentration effects may differently influence the fluorescence properties in PBS and DMF. However, this trend does not hold for **S-6**, where its quantum yield is higher in DMF than in PBS. This disparity suggests that the specific molecular structure and electronic properties of the α-glycoside lead to a different interaction with the organic solvent, possibly favoring its excited state stabilization and resulting in enhanced fluorescence emission. This phenomenon underscores the crucial role of solvent selection in modulating the photophysical properties of the investigated compounds. 

Previous studies, as reported in the literature [38,49], have demonstrated that the alkylation or introduction of a disaccharide at position 7 of **7-MC** significantly alters its fluorescence properties, resulting in increased emission. Similarly, our investigation shows that glycosylation of the thiol function by a monosaccharide also induces alterations in the fluorescence properties of **7-MC** in both aqueous buffer and DMF. The observed increase in fluorescence for S-glycosides compared to **7-MC** may be attributed to structural changes or a stabilization of the excited state of **7-MC** through favorable π-π interactions or hydrogen bonding.

A comparison of the fluorescence properties of **4-MUB**, **7-MC**, and *S*-glycosides across solvents provides insights into their structural and chemical properties. While **7-MC** exhibited low fluorescence characteristics in both DMF and PBS despite the addition of DTT, substitution of the thiol group into –OH or the introduction of a monosaccharide at position 7 led to solvent-dependent variations. This underscores the importance of considering both the molecular structure and solvent environment in understanding the fluorescence phenomena. 

Finally, in the scope of using those fluorescent compounds in a cellular context, the limit of detection (LOD) and limit of quantification (LOQ) for a representative *S*-glycoside, namely **S-3**, were determined in PBS buffer, and they were found to be, respectively, 0.2 µM and 0.05 µM (see Appendix A). Therefore, these fluorescent *S*-glycosyl coumarins are promising chemical probes for the detection of sub-micromolar-range processes.

## 3. Materials and Methods

### 3.1. Chemical Synthesis

*p*-Nitrophenyl-β-d-glucopyranoside, *p*-nitrophenyl-β-d-fucopyranoside, *p*-nitrophenyl-β-d-xylopyranoside, *p*-nitrophenyl-β-d-glucopyranosiduronic acid, *p*-nitrophenyl-β-d-galactopyranoside, and *p*-nitrophenyl-α-d-galactopyranoside were purchased from Carbosynth. **7-MC** and **4-MUB** were acquired from Sigma Aldrich. All chemical reagents were used directly without further purification. ^1^H and ^13^C NMR were recorded on Avance III HD NanoBay Bruker (Billerica, MA, USA) at 400 MHz (^13^C NMR: 100 MHz). Chemical shifts (δ) are reported in parts per million from tetramethylsilane (TMS) as the internal standard. Data are presented as follows: multiplicity (s = singlet; d = doublet; t = triplet; dd = doublet of doublet; m = multiplet; and b = broad), coupling constant (Hz), integration, and assignment. High-Resolution Mass Spectroscopy (HRMS) was performed on a Maxis Bruker UHR-Q-TOF spectrometer. Chromatographic analyses were conducted on a 1220 Infinity II LC system (Agilent Technologies, Les Ulis, France) equipped with the Infinity II Diode Array Detector (DAD). The mobile phase composition consisted of H_2_O + HCOOH 0.1% (solvent A) and CH_3_CN + HCOOH 0.1% (solvent B), and it was used according to the following elution gradient: 0–14 min, 90% A; 14–16 min, 40% A; 16–21 min, 0% A; and 21–25 min, 90% A. Semi-preparative HPLC purification was carried out on the previously described HPLC system, using the same buffer composition [17]. A Zorbax Eclipse XDB-C18 column (9.4 × 150 mm, 5 µ, Agilent Technologies Les Ulis, France) was employed at a flow rate of 2 mL/min, and the diode-array detector at 326 nm.

### 3.2. Enzymatic Thioglycosylation

General Procedure 1: As previously described in the literature [17,18], *p*-nitrophenyl-d-glucopyranoside (1.0 equiv., 50 mg, 10 mM) was dissolved in 25 mM citric acid/Na_2_HPO_4_ pH 6.0 buffer. Subsequently, 7-mercapto-4-methylcoumarin (2.5 equiv.), DTT (10 equiv.), MeOH (10–50%), and the thioglycoligase mutant (0.4–2.0 nM as final concentration) were added, and the mixture was left stirring at 37 °C overnight. The reaction was terminated by adding 50% of a quenching solution (CH_3_CN/HCOOH—10:1) and centrifuged at 12,000× *g* for 10 min to eliminate precipitated proteins. The supernatant was collected, concentrated under reduced pression, and purified by semi-preparative HPLC. 

4-Methylumbellifer-7-yl 1-thio-β-d-glucopyranosiduronic acid (**S-1**) [17,51,52] was obtained using *p*-nitrophenyl-β-d-glucopyranosiduronic acid (0.16 mmol, 1.0 equiv.) and the corresponding thioglycoligase DtGlcAE396Q (13.8 nmol, 8.6 × 10^−5^ equiv.) [17], according to General Procedure 1. The desired compound was obtained after purification as a white solid (32 mg, 54%). 

^1^H NMR (400 MHz, DMSO-*d*_6_) δ 7.67 (d, *J* = 8.4 Hz, 1H, H-5); 7.49–7.33 (m, 2H, H-6, H-8); 6.35 (bs, 1H, H-3); 4.97 (d, *J* = 9.8 Hz, 1H, H-1′); 3.75 (d, *J* = 8.5 Hz, H-5′); 3.40–3.24 (m, 2H, H-3′, H-4′); 3.15 (bt, *J* = 8.7 Hz, 1H, H-2′); 2.42 (s, 3H, -CH_3_). ^13^C NMR (100 MHz, DMSO-*d*_6_) δ 172.2 (1C, C-6′), 160.1 (1C, C-2), 153.6 (1C, C-4a), 153.5 (1C, C-8a), 141.1 (1C, C-4), 125.9 (1C, C-5), 124.7 (1C, C-6), 118.0 (1C, C-7), 115.8 (1C, C-8), 114.0 (1C, C-3), 86.2 (1C, C-1′), 79.0 (1C, C-5′), 78.1 (1C, C-4′), 72.6 (1C, C-2′), 72.0 (1C, C-3′), 18.5 (-CH_3_). HMRS (*m*/*z*) [M − H]^−^ calculated for C_16_H_17_O_8_S 369.0638; found: 369.0637. ${\left[\alpha \right]}_{D}^{20}$= −125.0° (*c* = 0.20 in MeOH).

4-Methylumbellifer-7-yl 1-thio-β-d-xylopyranoside (**S-2**) was obtained using *p*-nitrophenyl-β-d-xylopyranoside (0.18 mmol, 1.0 equiv.) and the corresponding thioglycoligase DtXylE161Q (7.6 nmol, 4.2 × 10^−5^ equiv.) [16], according to General Procedure 1. The desired compound was obtained after purification as a white powder (43 mg, 72%).

^1^H NMR (400 MHz, DMSO-*d*_6_) δ 7.70 (d, *J* = 8.3 Hz, 1H, H-5); 7.40–7.31 (m, 2H, H-8, H-6); 6.35 (s, 1H, H-3); 4.97 (d, *J* = 9.1 Hz, 1H, H-1′); 3.83 (dd, *J*_1_ = 10.5 Hz, *J*_2_ = 4.4 Hz, 1H, H_a_-5′); 3.33–3.21 (m, 3H, H-3′, H-4′, H_b_-5′); 3.16 (bt, *J* = 8.6 Hz, 1H, H-2′); 2.42 (s, 3H, -CH_3_). ^13^C NMR (100 MHz, DMSO-*d*_6_) δ 160.1 (1C, C-2), 153.5 (2C, C-4a, C-8a), 141.1 (1C, C-4), 126.0 (1C, C-5), 124.7 (1C, C-6), 118.0 (1C, C-7), 115.7 (1C, C-8), 114.0 (1C, C-3), 86.7 (1C, C-1′), 77.9 (1C, C-3′), 72.8 (1C, C-2′), 69.7 (1C, C-4′), 69.4 (1C, C-5′), 18.5 (1C, -CH_3_). HMRS (*m*/*z*) [M + H]^−^ calculated for C_15_H_17_O_6_S 325.0740; found: 325.0743. ${\left[\alpha \right]}_{D}^{20}$= −35.4° (*c* = 0.20 in CHCl_3_).

4-Methylumbellifer-7-yl 1-thio-β-d-glucopyranoside (**S-3**) [19,53,54,55] was obtained using *p*-nitrophenyl-β-d-glucopyranoside (0.17 mmol, 1.0 equiv.) and the corresponding thioglycoligase DtGlyE159Q (20.7 nmol, 12.2 × 10^−5^ equiv.) [4,18], according to General Procedure 1. The desired compound was obtained as white powder (37 mg, 63%).

^1^H NMR (400 MHz, DMSO-*d*_6_) δ 7.68 (d, *J* = 8.3 Hz, 1H, H-5); 7.42–7.29 (m, 2H, H-6, H-8); 6.29 (bd, *J* = 1.3 Hz, 1H, H-3); 4.80 (d, *J* = 9.7 Hz, 1H, H-1′); 3.72 (bd, *J* = 11.5 Hz, 1H, H_a_-6′); 3.50–3.31 (bd, *J* = 11.5 Hz, 1H, H_b_-6′); 3.29–3.16 (m, 2H, H-3′, H-5′); 3.15–2.96 (m, 2H, H-2′, H-4′); 2.37 (s, 3H, -CH_3_). ^13^C NMR (100 MHz, DMSO-*d*_6_) δ 159.6 (1C, C-2), 153.1 (1C, C-4a), 153.0 (1C, C-8a), 141.1 (1C, C-3), 125.4 (1C, C-5), 124.0 (1C, C-6), 117.3 (1C, C-7), 115.1 (1C, C-8), 113.4 (1C, C-4), 85.8 (1C, C-1′), 81.0 (1C, C-5′), 78.1 (1C, C-3′), 72.4 (1C, C-2′), 69.7 (1C, C-4′), 60.9 (1C, C-6′), 18.0 (-CH_3_). HMRS (*m*/*z*) [M + H]^−^ calculated for C_16_H_19_O_7_S 355.0846; found: 355.0846. ${\left[\alpha \right]}_{D}^{20}$= −78.5° (*c* = 0.30 in MeOH).

4-Methylumbellifer-7-yl 1-thio-β-d-galactopyranoside (**S-4**) [20] was obtained using *p*-nitrophenyl-β-d-galactopyranoside (0.17 mmol, 1.0 equiv.) and the corresponding thioglycoligase DtGlyE159Q (20.7 nmol, 12.2 × 10^−5^ equiv.), according to General Procedure 1. The desired compound was obtained as white powder (32 mg, 55%).

^1^H NMR (400 MHz, DMSO-*d*_6_) δ 7.67 (d, *J* = 8.3 Hz, 1H, H-5); 7.44 (s, 1H, H-8); 7.37 (bd, *J* = 8.3 Hz, 1H, H-6); 6.33 (s, 1H, H-3); 4.80 (d, *J* = 9.6 Hz, 1H, H-1′); 3.69–3.67 (m, 1H, H-sugar); 3.56–3.41 (m, 5H, H-sugar); 3.36–3.29 (m, 4H, H-sugar); 2.42 (s, 3H, -CH_3_). ^13^C NMR (100 MHz, DMSO-*d*_6_) δ 160.2 (1C, C-2), 153.6 (1C, C-4a), 153.5 (1C, C-8a), 142.0 (1C, C-3), 125.9 (1C, C-5), 124.0 (1C, C-6),115.3 (1C, C-8), 113.8 (1C, C-3), 86.8 (1C, C-1′), 79.8 (1C, C-2′), 75.1 (1C, C-sugar), 69.5 (1C, C-sugar), 68.8 (1C, C-sugar), 61.0 (1C, C-6′), 18.5 (1C, -CH_3_). HMRS (*m*/*z*) [M + Na]^+^ calculated for C_16_H_18_O_7_S 377.0673; found: 377.0672. ${\left[\alpha \right]}_{D}^{20}$= −55.2° (*c* = 0.30 in CHCl_3_).

4-Methylumbellifer-7-yl 1-thio-β-d-fucopyranoside (**S-5**) was obtained using *p*-nitrophenyl-β-d-fucopyranoside (0.18 mmol, 1.0 equiv.) and the corresponding DtGlyE159Q (23.2 nmol, 12.9 × 10^−5^ equiv.), according to General Procedure 1. The desired compound was obtained as white powder (40 mg, 68%).

^1^H NMR (400 MHz, DMSO-*d*_6_) δ 7.69 (d, *J* = 8.4 Hz, 1H, H-5); 7.41 (s, 1H, H-8); 7.36 (d, *J* = 8.4 Hz, 1H, H-6); 6.34 (s, 1H, H-3); 4.81 (d, *J* = 9.4 Hz, 1H, H-1′); 3.77 (q, *J* = 6.3 Hz, 1H, H-5′); 3.60–3.39 (m, 3H, H-3′, H-4′, H-2′); 2.42 (s, 3H, -CH_3_); 1.15 (d, *J* = 6.3 Hz, -CH_3_ sugar). ^13^C NMR (100 MHz, DMSO-*d*_6_) δ 125.9 (1C, C-5), 124.3 (1C, C-6), 115.2 (1C, C-8), 113.8 (1C, C-3), 86.3 (1C, C-1′), 75.2 (1C, C-sugar), 74.6 (1C, C-5′), 71.7 (1C, C-sugar), 69.1 (1C, C-2′), 18.5 (1C, -CH_3_), 17.3 (1C, -CH_3_ sugar). HMRS (*m*/*z*) [M + H]^−^ calculated for C_16_H_17_O_6_S 325.0740; found: 325.0743. ${\left[\alpha \right]}_{D}^{20}$= −93.4° (*c* = 0.20 in CHCl_3_).

4-Methylumbellifer-7-yl 1-thio-α-d-galactopyranoside (**S-6**) was obtained using *p*-nitrophenyl-α-d-galactopyranoside (0.17 mmol, 1.0 equiv.) and the corresponding DtαGalD546A (32.8 nmol, 19.3 × 10^−5^ equiv.), according to General Procedure 1. The desired compound was obtained as white powder (37 mg, 63%).

^1^H NMR (400 MHz, DMSO-*d*_6_) δ 7.66 (d, *J* = 8.4 Hz, 1H, H-5); 7.49 (s, 1H, H-8); 7.42 (d, *J* = 8.4 Hz, 1H, H-6); 6.33 (s, 1H, H-3); 5.83 (d, *J* = 5.4 Hz, 1H, H-1′); 4.05 (dd, *J*_1_ = 10.0 Hz, *J*_2_ = 5.4 Hz, 1H, H-2′); 3.98 (t, *J* = 6.2 Hz, 1H, H-5′); 3.79 (bd, *J* = 3.2 Hz, 1H, H-4′); 3.56 (dd, *J*_1_ = 10.9 Hz, *J*_2_ = 6.2 Hz, 1H, H_a_-6′); 3.39 (dd, *J*_1_ = 10.9 Hz, *J*_2_ = 6.2, H_b_-6′); 2.41 (s, 3H, -CH_3_). ^13^C NMR (100 MHz, DMSO-*d*_6_) δ 160.2 (1C, C-2), 153.6 (1C, C-4a), 153.5 (1C, C-8a), 142.2 (1C, C-4), 125.7 (1C, C-5), 125.6 (1C, C-6), 117.8 (1C, C-7), 116.6 (1C, C-8), 113.9 (1C, C-3), 88.7 (1C, C-1′), 73.4 (1C, C-5′), 71.1 (1C, C-3′), 68.7 (1C, C-4′), 68.3 (1C, C-2′), 60.6 (1C, C-6), 18.5 (1C, -CH_3_). HMRS (*m*/*z*) [M + H]^−^ calculated for C_16_H_19_O_7_S 355.0846; found: 355.0842. ${\left[\alpha \right]}_{D}^{20}$= +190.7° (*c* = 0.30 in CHCl_3_).

### 3.3. Protein Expression and Production

Production and purification of the thioglycoligase mutants were performed as previously described by Guillotin et al. [6]. Briefly, *Escherichia coli* Rosetta (DE3), transformed with expression plasmid containing thioglycoligase coding genes, were grown in LB medium, supplemented with the corresponding antibiotics, Chloramphenicol (34 μg/mL) and Kanamycin (30 μg/mL), at 37 °C until OD_600_ reached 0.4–0.6. Induction was then performed by adding 1 mM IPTG, and the culture was incubated overnight at 25 °C, while shaking at 220 rpm. Cells were harvested, lysed by freeze–thaw cycles, and sonicated. The supernatant was clarified by heating at 70 °C for 15 min, before centrifugation. Finally, the lysate was loaded onto a Ni^2+^-NTA column (HisPure, Thermo Fisher, Waltham, MA, USA) and purified by elution with buffer containing 500 mM imidazole. The concentration of the various fractions collected was determined by Bradford assay [56].

### 3.4. Cellular Assays

Cell proliferation of five distinct cell lines was evaluated using the colorimetric MTT (thiazolyl blue tetrazolium bromide, Sigma) assay. Human keratinocyte normal skin HaCaT was cultivated in DMEM supplemented with 4.5 g/L glucose, 2 mM L-glutamine, and 10% FBS. Cancer cell lines and their respective growth media were obtained from CLS Cell Line Service GmbH (Eppelheim, Germany). Specifically, human skin melanoma SK-MEL-28 and human brain glioma HS683 were cultivated in DMEM with identical supplements as HaCaT cells. The human lung carcinoma cell line A-549 was grown in DMEM/Ham’s F12 (1:1) medium supplemented with 2 mM L-glutamine and 5% FBS. Additionally, human breast adenoma carcinoma MCF-7 cells were sustained in EMEM supplemented with 2 mM l-glutamine, sodium pyruvate, non-essential amino acids (NEAAs), 10 µg/mL insulin human, and 10% FBS. The MTT assay relies on the reduction of the yellow product thiazolyl blue tetrazolium bromide (MTT) to formazan, a purple-blue product, by mitochondrial dehydrogenases in metabolically active cells. The number of living cells after incubation in the presence (or absence, control) of the tested molecules is directly proportional to the blue color, which was measured by spectrophotometry. To perform the assay, cells were initially seeded at a density of 2500 cells/mL in 96-well culture plates (Nunc™ Edge 2.0, Fisher) and allowed to adhere and proliferate over a 24-hour incubation period. Following this, the cells were treated with various concentrations of the tested compounds, which were previously prepared as serial dilutions and dissolved in 0.1% DMSO. These stable solutions guarantee the consistency of experimental conditions over extended periods. Each concentration of the tested compounds underwent triplicate testing (*n* = 3), and cells were subsequently incubated for 72 h to allow for thorough evaluation. Following this incubation period, MTT reagent (5 mg/mL solution in PBS) was introduced to each well (10% *v*/*v*), and cells were further incubated for 4 h. Then, the culture medium was carefully removed, and the resulting blue formazan crystals were dissolved in 100 µL of 100% DMSO. Absorbance readings were taken at 540 nm, referencing a wavelength of 620 nm, using a spectrophotometer. In determining the IC_50_, the absorbance of serial dilutions of each cell line, treated under identical conditions but without the presence of the tested compounds, was measured. This procedure facilitated the construction of a standard curve, from which the IC_50_ value, denoting the concentration at which cell growth is reduced by 50%, was determined.

### 3.5. Optical Spectroscopy

Both absorbance and fluorescence spectra were measured on a Tecan INFINITE M PLEX 200 Pro plate reader equipped with monochromator allowing a scan from 230 to 1000 nm. All synthesized molecules were initially dissolved in 10 mM DMF and subsequently diluted in freshly prepared PBS (pH = 7.4) to achieve a final concentration of 0.1 mM. The solutions were then transferred to a Corning (Corning, NY, USA) UV-transparent 96-well plate. Absorbance spectra were recorded from 230 to 800 nm. Emission fluorescence spectra were recorded from 320 to 800 nm, under excitation at 280 nm. Excitation fluorescence spectra were recorded from 230 to 380 nm for an emission at 420 nm. All spectra were recorded with a wavelength step size of 2 nm. Data processing was performed with Excel software 2016. Fluorescence spectra are presented as normalized spectra ranging from 0 to 1. The relative quantum yields were calculated according to Equation (1) [57]:(1)$$\Phi_{T}=(A_{R}/A_{T})\times (F_{T}/F_{R})\times (n_{T}/n_{R})\times \Phi_{R}$$
where Φ is the fluorescence quantum yield, *A* is the absorbance at the excitation wavelength, *F* is the area under the corrected emission curve, and *n* is the refractive index of the solvent for the test compound (*T*) and the reference (*R*). The 4-MUB value in DMF was used as the reference [50]. The LOD (resp. LOQ) values were determined as the minimal concentration of product **S-3** required to yield a fluorescence that was twice greater than (respectively, 5-times that of) the fluorescence noise on a sample containing PBS, using an excitation wavelength of 280 nm and an emission wavelength of 386 nm (Appendix A).

## 4. Conclusions

In summary, this study highlights the proficiency of mutated GHs in catalyzing the thioligation reaction between activated sugar substrates and **7-MC**, leading to the efficient synthesis of *S*-glycosides. This enzymatic approach circumvents the need for laborious protection/deprotection protocols and purification steps commonly encountered in chemical synthesis, thereby offering a more efficient and convenient method for obtaining *S*-glycosides with intended configurations and in an eco-responsible way. The synthetized derivatives were produced in good yields, displaying the efficiency and versatility of the biocatalytic approach. Furthermore, the cytotoxicity assays conducted on multiple cancer cell lines, alongside a healthy cell line, revealed promising non-cytotoxic activities for most *S*-derivatives. These findings suggest their potential utility as chemical probes in a cellular context, with the carbohydrate moiety increasing the solubility of the aglycon and owing targeting of specific lectins [17,18]. Moreover, fluorescence studies revealed a dramatic increase in emission for *S*-glycosides compared to **7-MC**, indicating potential applications in fluorescence-based assays and imaging techniques. This fluorescence enhancement paves the way for the development of sensitive detection methods and imaging probes. Overall, these results emphasize the utility of enzymatic synthesis in producing bioactive *S*-derivatives and offer valuable insights into their cytotoxic and fluorescence properties.

## Data Availability

Data are contained within the article or Appendix A.

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
