# Peer review of "Biocatalytic Synthesis of Coumarin S-Glycosides: Towards Non-Cytotoxic Probes for Biomedical Imaging and Sensing"

_molecules, 2024, doi:10.3390/molecules29061322_

Round 1
Reviewer 1 Report
Comments and Suggestions for Authors
See the review file attached
Author Response
The authors wish to thank the reviewer for these useful comments.
1. The manuscript describes an efficient protocol for enzymatic synthesis of 7-mercapto-4-methylcoumarin
S-glycosides of a series of monosaccharides using several genetically modified glycosyl
hydrolases with lowered hydrolytic activity that are capable of acting as thioglycoligases. The
thioglycosides obtained possess good fluorescence properties exceeding those of the parent 7-mercapto-
4-methyl-coumarin and are devoid of cytotoxicity against various human cell lines, which make promise
for future synthesis of similar fluorescent thioglycosides of more complex carbohydrates that could be
useful for diverse biomedical applications.
- RECOMMENDATION. In Reviewer's opinion, the manuscript should definitely be published after
correction of the following issues.
- REFERENCES. The authors should add the Ref. 57 (it is empty now).
The reference was reformulated as follows: "48. Abu-Eittah, R.H.; El-Tawil, B.A.H. The Electronic Absorption Spectra of Some Coumarins. A Molecular Orbital Treatment. Can. J. Chem. 1985, 63, 1173–1179, doi:10.1139/v85-200.”
- MANUSCRIPT. The manuscript contains some misprints and errors. The authors should double check
the text of the manuscript for errors. Some examples are listed below:
4.1. Page 1 (lines 27-28): it is incorrect to state that “O-glycosides significantly contribute to the structural
complexity and functionality” of “glycoproteins and glycolipids” since O-glycosides are NOT components
of the latter which are complex molecules that contain oligosaccharide chains made of several
monosaccharide residues linked by O-glycosidic linkages. The sentence should be reformulated.
The sentence was reformulated as follows: “ Abundantly present in natural products and plant secondary metabolites O-glycosides significantly contribute to the structural complexity and functionality of these bioactive molecules. “
4.2. Page 1, line 43: metallic catalysis reactions --> metal catalyzed reactions
Done.
4.3. The authors should correct the names of ALL compounds since they do not follow the carbohydrate
nomenclature rules. All aglycons should be separated by space from glycon. For example, page 3, lines
100-101: p-nitrophenyl-D-glycopyranosides --> p-nitrophenyl D-glycopyranosides.
Done.
4.4. Page 3, line 113: purely --> pure
Done.
4.5. Page 5, lines 175-177: the authors claim that they measured spectra in ”freshly prepared phosphatebuffered
saline (PBS) at pH = 7.4, to mimick physiological conditions”. However, the Experimental (page
10, lines 398-399) contains different information – “all synthesized molecules were dissolved in DMF and
diluted in freshly prepared PBS 1x (pH = 7.4) up to 0.1 mM” that is the DMF–PBS mixture was used rather
than pure PBS, which suggests that the prepared S-glycosides have limited aqueous solubility. The authors
should clearly indicate the composition of the solvent mixture used including the amount of DMF used
for dissolution of compounds. Additionally, the authors should address the solubility issues regarding
future biological applications.
This was corrected as following: “ UV-visible spectral analysis was conducted using dimethylformamide (DMF) as the initial solvent at a concentration of 10 mM to ensure optimal solubility of the compounds. Subsequently, the analysis was carried out under two distinct conditions: firstly, in DMF at a final concentration of 0.1 mM, and secondly, in a mixture PBS:DMF (99:1%) at the same final concentration of 0.1 mM. These two experimental setups provided valuable insights into the spectral characteristics of the compounds in both a pure organic solvent and a solvent system closely resembling physiological conditions. “
4.6. Page 5, line 185: increase in solvent polarity induced shifts --> increase in solvent polarity (from DMF
to PBS) induced shifts
This was corrected as following: “Furthermore, an increase in solvent polarity, transitioning from DMF to PBS “
4.7. Page 5, lines 182: extinction coefficients --> extinction coefficients (in DMF). The authors should
indicate that in PBS the situation is completely different and extinction coefficients for some
thioglycosides are lower than that for the parent thiol.
This was corrected as following: “ Similarly, 7-MC and S-1 to S-6 compounds also absorb in this UV range (lmax=328–330 nm). However, the extinction coefficients for S-glycosides exhibit variability. In the organic solvent, these values are found to be higher compared to those calculated for 4-MUB and 7-MC. Conversely, in PBS the extinction coefficients for S-glycosides are lower compared to the parent thiol 7-MC. This highlights the profound impact of glycosylation on absorption properties, as well as the importance of solvent choice. “
4.8. Page 5, line 192: 320-333 nm --> 320–334 nm
Done.
4.9. Page 6, lines 219-221: The authors claim that “DMF … facilitates … stabilization of the excited state,
thereby resulting in heightened fluorescence emission” when compared with PBS. In fact, this is correct
only for compound S-6, the only one featured by alpha-configuration. In all other cases quantum yields
are higher in PBS when compared with those in DMF. The text should be modified accordingly.
This was corrected as following: “Notably, when comparing their behavior in PBS to DMF, we observed a consistent increase in emission intensity for the S-derivatives in PBS, except for S-2 and S-6. This discrepancy primarily arises from the different polarity and hydrogen bonding capabilities of the two solvents. PBS is renowned for its high polarity and the presence of phosphate ions, which can readily engage in hydrogen bonding interactions with the functional groups of the S-glycosides. These interactions facilitate the stabilization of the excited state, consequently leading to heightened fluorescence emission. Moreover, the polar environment of PBS may shield the molecules from non-radiative decay processes, further enhancing fluorescence intensity. Conversely, the organic nature of DMF introduces distinct dynamics to the excited state relaxation pathways of the S-glycosides. Being a less polar solvent, DMF may exert a weaker influence on the stabilization of the excited states compared to PBS. Additionally, the formation of hydrogen bonding interactions, different from those present in PBS, might alter the energetic landscape of the excited states, potentially resulting in lower fluorescence emission and quantum yields. Furthermore, microenvironment effects within each solvent could also contribute to the observed differences in fluorescence behavior. Variations in solute-solvent interactions, solvent accessibility to the chromophore and local concentration effects may differently influence the fluorescence properties in PBS and DMF. However, this trend does not hold for S-6, where its quantum yield is higher in DMF than in PBS. This disparity suggests that the specific molecular structure and electronic properties of the a-glycoside lead to a different interaction with the organic solvent, possibly favoring its excited state stabilization and resulting in enhanced fluorescence emission. This phenomenon underscores the crucial role of solvent selection in modulating the photophysical properties of the investigated compounds.”
4.10. Page 7, line 246: Please double check if D-isomer of p-nitrophenyl beta-D-fucopyranoside was indeed
used (L-isomer is more common).
p-nitrophenyl beta-D-fucopyranoside was effectively used.
4.11. Page 7, line 247: glucuronic acid--> glucopyranosiduronic acid
Done.
4.12. Page 7, line 249: was--> were
Done.
4.13. Page 7, line 264: detector recorded the UV spectrum at 326 nm --> detector at 326 nm
Done.
4.14. Page 7, line 278: glucuronic acid--> glucopyranosiduronic acid
Done.
4.15. Page 8, line 291: xylopyranose --> xylopyranoside
Done.
4.16. Page 9, line 347: J2 = XXX --> please indicate the J value
Done.
4.17. Page 9, line 334: 1.15 (s, 3H, -CH3 sugar) --> 1.15 (d, J = 6.3 Hz, 3H, H-6’). The coupling constants
MUST be equal. Please double check your data. See also the line 333.
Done.
4.18. For ALL compounds synthesized specific optical rotation values MUST be given. These values are
compulsory for adequate characterization of chiral compounds.
Done.
- FUNDING INFORMATION. There is conflicting (possibly misleading) information concerning “external
funding” in the FUNDING and ACKNOWLEDGMENTS sections. The authors should double check this
information.
These have been checked.
Reviewer 2 Report
Comments and Suggestions for Authors
The core of the manuscript submitted by Burrini et al is the enzymatic synthesis of some coumarin S-glycosides (S1-S6). From the point of view of novelty, the research groups herein involved have applied the same enzymatic approach to obtain S1 (ref. 17); moreover, S1 and S3 have been obtained through other enzymatic approaches (refs. 19, 60, 61). However, the present manuscript also reports experiments showing non-cytotoxic behavior of most of the obtained products; moreover, the fluorescence properties of the products have been assessed, so that on the whole, the potential utility of them as chemical probes in a cellular context is suggested.
I have the following comments:
1- a) In the caption to Figures S25 to S27 (Supplementary Material, page 1) it is stated that S3, S5 and S6 are undescribed compounds. However, S3 is reported in ref. 19 (page 6, Table 1 (b), Fig. 2d). In contrast, in the section of enzymatic thioglycosylation (manuscript, pages 7-9) references are provided for S1, S3 and S4, but not for S2, S5 and S6. ¿Is S2 a not described compound? Please check these compounds.
b) ¿Have S2 (if not a novel compound) and S4 already been prepared enzymatically? If not, the novelty should also be stressed in the paper.
c) The synthesis of S3 through the enzymatic approach in ref. 19 affords 30 %, a much lower yield compared with 63 % (submitted manuscript). This should be remarked to highlight the usefulness of the herein reported approach.
2- “Conversion rate” (lines 117, 131 and third column of Table 1; please revise the whole manuscript): the reported values are just conversions but not rates, so that the term is to be replaced by “conversion”.
3- a) 1H-NMR spectrum of S4: the integration of the Hs in the sugar moiety, reported between 3.69 and 3.29 ppm (lines 321 and 322), does not match with the number of involved Hs. Moreover, such Hs should be assigned as done with the corresponding Hs in S1-S6.
b) 1H-NMR spectrum of S6: please provide the J depicted as XXX (line 347).
4- To illustrate the usefulness of 7-MC as Raman reporter, 12 references have been provided (refs. 27-38). This is excessive for a paper which is not a review; moreover, on the whole, 66 references have been given, which seem to be excessive for an article-type manuscript. Please revise refs. 27-38 and choose 2-3 representative works.
5- Formal remarks:
a) Please revise the list of references checking reference by reference. In some cases, the abbreviated name of the journal is given; in others, full name. The same applies to number pages (sometimes only the first is provided, sometimes both first and last). Moreover, in most cases the name of the microorganisms is not written in italics. Please write “D” in capital letter when it is part of the name of a carbohydrate. In ref. 62, title appears in full capital letters. No journal name is provided for ref. 57.
b) In Tables 2 and 3, replace footnames below the table (they have been presented as part of the title).
c) Please revise that “S” always stays in italics (for instance, in lines 218, 420, 427, it stays in normal letter).
d) Following names are not to be written in capital letters: “5-Fluorouracil” (line 149), “Etoposide” (line 149).
Comments on the Quality of English Language
English is fine. Only a remark is done: “7-MC and 4-MUB was acquired…” (line 249), please replace “was” by “were”.
Author Response
The authors wish to thank the reviewer for these useful comments.
The core of the manuscript submitted by Burrini et al is the enzymatic synthesis of some coumarin S-glycosides (S1-S6). From the point of view of novelty, the research groups herein involved have applied the same enzymatic approach to obtain S1 (ref. 17); moreover, S1 and S3 have been obtained through other enzymatic approaches (refs. 19, 60, 61). However, the present manuscript also reports experiments showing non-cytotoxic behavior of most of the obtained products; moreover, the fluorescence properties of the products have been assessed, so that on the whole, the potential utility of them as chemical probes in a cellular context is suggested.
I have the following comments:
1- a) In the caption to Figures S25 to S27 (Supplementary Material, page 1) it is stated that S3, S5 and S6 are undescribed compounds. However, S3 is reported in ref. 19 (page 6, Table 1 (b), Fig. 2d). In contrast, in the section of enzymatic thioglycosylation (manuscript, pages 7-9) references are provided for S1, S3 and S4, but not for S2, S5 and S6. ¿Is S2 a not described compound? Please check these compounds.
This was corrected as follows: 1H and 13C NMR of undescribed compounds S-2, S-5 and S-6.
- b) ¿Have S2 (if not a novel compound) and S4 already been prepared enzymatically? If not, the novelty should also be stressed in the paper.
This was corrected as follows: “Notably, our work marks the inaugural enzymatic synthesis of S-2, S-4 to S-6, a significant milestone in glycoscience. This innovative methodology not only facilitates precise control over stereochemistry and molecular structure, but also represents a pioneering leap forward in the field.”
- c) The synthesis of S3 through the enzymatic approach in ref. 19 affords 30 %, a much lower yield compared with 63 % (submitted manuscript). This should be remarked to highlight the usefulness of the herein reported approach.
This was corrected as follows: “Our research underscores the potential of enzymatic glycosylation as a robust method for synthetizing S-derivatives with remarkable efficiency [19].”
2- “Conversion rate” (lines 117, 131 and third column of Table 1; please revise the whole manuscript): the reported values are just conversions but not rates, so that the term is to be replaced by “conversion”.
Done.
3- a) 1H-NMR spectrum of S4: the integration of the Hs in the sugar moiety, reported between 3.69 and 3.29 ppm (lines 321 and 322), does not match with the number of involved Hs. Moreover, such Hs should be assigned as done with the corresponding Hs in S1-S6.
The characterization of such compound corresponds to the H-NMR spectrum in the following reference: “20. Rodrigue, J.; Ganne, G.; Blanchard, B.; Saucier, C.; Giguère, D.; Shiao, T.C.; Varrot, A.; Imberty, A.; Roy, R. Aromatic Thioglycoside Inhibitors against the Virulence Factor LecA from Pseudomonas Aeruginosa. Org. Biomol. Chem. 2013, 11, 6906–6918, doi:10.1039/C3OB41422A.”
- b) 1H-NMR spectrum of S6: please provide the J depicted as XXX (line 347).
Done.
4- To illustrate the usefulness of 7-MC as Raman reporter, 12 references have been provided (refs. 27-38). This is excessive for a paper which is not a review; moreover, on the whole, 66 references have been given, which seem to be excessive for an article-type manuscript. Please revise refs. 27-38 and choose 2-3 representative works.
Done.
5- Formal remarks:
- a) Please revise the list of references checking reference by reference. In some cases, the abbreviated name of the journal is given; in others, full name. The same applies to number pages (sometimes only the first is provided, sometimes both first and last). Moreover, in most cases the name of the microorganisms is not written in italics. Please write “D” in capital letter when it is part of the name of a carbohydrate. In ref. 62, title appears in full capital letters. No journal name is provided for ref. 57.
Done.
- b) In Tables 2 and 3, replace footnames below the table (they have been presented as part of the title).
Done.
- c) Please revise that “S” always stays in italics (for instance, in lines 218, 420, 427, it stays in normal letter).
Done.
- d) Following names are not to be written in capital letters: “5-Fluorouracil” (line 149), “Etoposide” (line 149).
Done.
Reviewer 3 Report
Comments and Suggestions for Authors
In this manuscript, Lafite, Daniellou and co-workers report the enzymatic synthesis of 6 coumarin S-glycosides using thioglycoligases which were previously developed by the same group. In this work, they incorporated coumarin, a fluorophore, with different glycosides, yielding non-toxic fluorescent S-glycosides. They did some optical study of the resulting compounds. I will only consider it for acceptance after major revisions.
1. The authors gave a very good introduction to the enzymatic S-glycosylation. It would be easier for the readers to follow if the authors could add some graphical figures to describe the evolution of their enzymes.
2. Major concern: The authors claimed that this reaction methodology is very efficient. However, in the substrate scope, the authors only gave 6 examples which is not enough to support their conclusion. More sugars, different activation groups (pNP), different protecting groups on hydroxyl groups and different substitutions on coumarin should be explored.
3. The authors should mention the α/β selectivity of the glycosylation in table 1.
4. Major concern: In the evaluation of the resulting compounds, 7-methoxy-4-methylcoumarin or 7- methylthio-4-methylcoumarin should be used as the truncated substructures rather than the 4-MUB or 7-MC. The authors should prove the benefits of combining sugars with coumarin. From the current version, I did not see any necessity.
5. The authors conducted fluorescence evaluation in PBS. More buffers of different pH should be explored.
6. Major concern: Cell staining should be pursued to see the distribution of the compounds in cellular environment.
Author Response
The authors wish to thank the reviewer for the comments.
In this manuscript, Lafite, Daniellou and co-workers report the enzymatic synthesis of 6 coumarin S-glycosides using thioglycoligases which were previously developed by the same group. In this work, they incorporated coumarin, a fluorophore, with different glycosides, yielding non-toxic fluorescent S-glycosides. They did some optical study of the resulting compounds. I will only consider it for acceptance after major revisions.
- The authors gave a very good introduction to the enzymatic S-glycosylation. It would be easier for the readers to follow if the authors could add some graphical figures to describe the evolution of their enzymes.
Thank you very much for this positive comment. As far as the design of enzymes is concerned, please refer to our previous papers, so to circumvent self-plagiarism.
- Major concern: The authors claimed that this reaction methodology is very efficient. However, in the substrate scope, the authors only gave 6 examples which is not enough to support their conclusion. More sugars, different activation groups (pNP), different protecting groups on hydroxyl groups and different substitutions on coumarin should be explored.
The methodology is efficient in term of reaction yield/ conversion yield, and quite versatile as noticed by the reviewer (6 examples). The exploration of the substrate scope is highly dependent of the nature of the different enzyme and is not the purpose of this study.
- The authors should mention the α/β selectivity of the glycosylation in table 1.
Done.
- Major concern: In the evaluation of the resulting compounds, 7-methoxy-4-methylcoumarin or 7-methylthio-4-methylcoumarin should be used as the truncated substructures rather than the 4-MUB or 7-MC. The authors should prove the benefits of combining sugars with coumarin. From the current version, I did not see any necessity.
In this article they have already demonstrated the increase of fluorescence once that the 7-MC is glycosylated (in this case was a cellobiose) 38. Barr, B.K.; Holewinski, R.J. 4-Methyl-7-Thioumbelliferyl-β-d-Cellobioside: A Fluorescent, Nonhydrolyzable Substrate Analogue for Cellulases. Biochemistry 2002, 41, 4447–4452, doi:10.1021/bi015854q.
This is also the case in our study and was already depicted in the conclusion. In addition, the carbohydrate moity increases the solubility and allows the targeting of specific lectins on cells.
- The authors conducted fluorescence evaluation in PBS. More buffers of different pH should be explored.
In this article several organic solvents were tested to evaluate the fluorescence properties of 7-MC and alkylated derivative. 49. Lanterna, A.E.; González-Béjar, M.; Frenette, M.; Scaiano, J.C. Photophysics of 7-Mercapto-4-Methylcoumarin and Derivatives: Complementary Fluorescence Behaviour to 7-Hydroxycoumarins. Photochem. Photobiol. Sci. 2017, 16, 1284–1289, doi:10.1039/C7PP00121E. For 4-MUB there are several articles in literature (D. Jacquemin et al. Time dependent density functional theory investigation of the absorption, fluorescence, and phosphorescence spectra of solvated coumarins) where they have evaluated the fluorescence properties of the compound in different conditions. In our study the goal was to use a buffer closes to the cellular environment.
- Major concern: Cell staining should be pursued to see the distribution of the compounds in cellular environment.
Thank you. Indeed, such study will require too many experiments to be performed and will be highly dependent on the cellular types. Therefore, they will be reported in due time in a following paper.
Round 2
Reviewer 1 Report
Comments and Suggestions for Authors
The authors resolved all issues. The manuscript should be published.
Author Response
Thank you very much.
Reviewer 2 Report
Comments and Suggestions for Authors
The manuscript is now improved according to the provided comments.
Author Response
Thank you very much.
Reviewer 3 Report
Comments and Suggestions for Authors
As the author mentioned "In addition, the carbohydrate moity increases the solubility and allows the targeting of specific lectins on cells." in the rebutta. These benefits should be emphasized in the discussion or conclusion parts and add some suitable references.
Author Response
As the author mentioned "In addition, the carbohydrate moity increases the solubility and allows the targeting of specific lectins on cells." in the rebutta. These benefits should be emphasized in the discussion or conclusion parts and add some suitable references.
Thank you very much for this suggestion. We therefore added in the conclusion page 15 line 469 : "These findings suggest their potential utility as chemical probes in a cellular context, with the carbohydrate moiety increasing the solubility of the aglycon and owing targeting of specific lectins [17-18]."